# BASISVAE: ORTHOGONAL LATENT SPACE FOR DEEP DISENTANGLED REPRESENTATION

## ABSTRACT

The variational autoencoder, one of the generative models, defines the latent space for the data representation, and uses variational inference to infer the posterior probability. Several methods have been devised to disentangle the latent space for controlling the generative model easily. However, due to the excessive constraints, the more disentangled the latent space is, the lower quality the generative model has. A disentangled generative model would allocate a single feature of the generated data to the only single latent variable. In this paper, we propose a method to decompose the latent space into basis, and reconstruct it by linear combination of the latent bases. The proposed model called BasisVAE consists of the encoder that extracts the features of data and estimates the coefficients for linear combination of the latent bases, and the decoder that reconstructs the data with the combined latent bases. In this method, a single latent basis is subject to change in a single generative factor, and relatively invariant to the changes in other factors. It maintains the performance while relaxing the constraint for disentanglement on a basis, as we no longer need to decompose latent space on a standard basis. Experiments on the well-known benchmark datasets of MNIST, 3DFaces and CelebA demonstrate the efficacy of the proposed method, compared to other state-of-the-art methods. The proposed model not only defines the latent space to be separated by the generative factors, but also shows the better quality of the generated and reconstructed images. The disentangled representation is verified with the generated images and the simple classifier trained on the output of the encoder.

## 1 INTRODUCTION

The proper choice of data representation is highly correlated with the difficulty of task learning for a given machine learning approach (Higgins et al., 2017; Kim et al., 2018; Kim & Cho, 2018; 2019). Using a representation appropriate to specific task and data domain can significantly improve the robustness and successful learning of the model (Kim et al., 2018; Kim & Cho, 2018; 2019; Bengio et al., 2013). In particular, disentangled representation is useful when dealing with data with various features, and can be effective for a large variety of domains and tasks (Bengio et al., 2013; Ridgeway, 2016). A latent space is disentangled if single latent units are subject to changes in single generative factors, and relatively invariant to changes in other factors (Bengio et al., 2013). For example, a generative model trained on a dataset of facial images learns independent latent units subject to single independent generative factors such as hair color, gender, and emotion. We define the disentangled representation using equation (1). A change of single generative factor is consistent to the change of single coefficient $c_i$, but not to $c_j$ for $i \neq j$.

$$\mathbf{z} = \Sigma_i c_i \mathbf{e_i} \tag{1}$$

where $\mathbf{z} \in \mathbb{R}^Z$ is a latent variable, $\mathbf{e_i} \in \mathbb{R}^Z$ is a standard unit vector, and $c_i \in \mathbb{R}$ is a coefficient. Disentangled representation can be useful in several machine learning tasks including transfer learning and zero-shot learning (Lake et al., 2017). Moreover, unlike most representation learning algorithms, disentangled representation can be interpreted because they are consistent with the variability of the data (Dupont, 2018). The variational autoencoder (VAE) is used to define the latent space by approximating the posterior distribution with approximation as follows (Kingma & Welling, 2013).

$$\log p_\theta(x) \geq -\mathcal{D}_{KL}[q_\phi(z|x) \| p_\theta(z)] + \mathbb{E}_{q_\phi(z|x)}[\log p_\theta(x|z)] \tag{2}$$

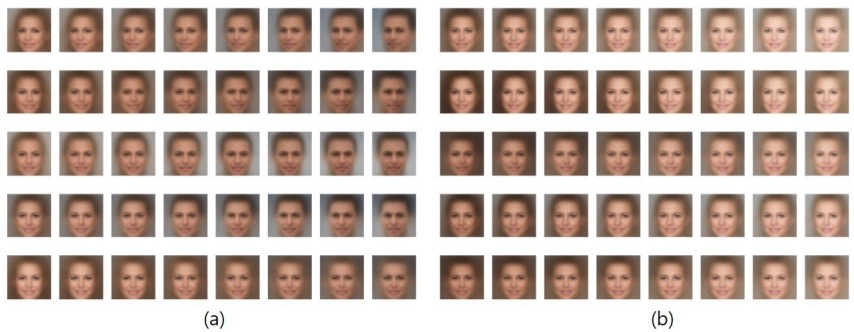

(a)                                                                                    (b)

Figure 1: In the vanilla VAE model, an interpolation experiment results in only changing the single generative factor only for (a) gender and (b) skin color.

where $\mathbb{D}_{K}L$ is Kullback-Leibler divergence, $q_{\phi}(z|x)$ is a posterior distribution inferred by encoder, $p_{\theta}(z)$ is a prior distribution, and $p(x|z)$ is a likelihood or decoder. Since the VAEs are powerful to define the latent space, it is often used for disentangled representation learning (Higgins et al., 2017; Dupont, 2018). However, in most cases, the quality of the generated data is relatively low because of the added constraints to the loss function (Kim & Cho, 2019). This is because the scale of the latent variable to represent the generative factor drops from $\mathbb{R}^{Z}$ to $\mathbb{R}^{1}$ when the generative model is $f :$ $\mathcal{Z} \subset \mathbb{R}^{Z} \to \mathcal{X} \subset \mathbb{R}^{X}$, where $Z$ and $X$ are the dimensions of the latent space and data, respectively. Several researchers have studied for disentanglement, but the trade-off with performance has not been considered (Higgins et al., 2017; Dupont, 2018; Chen et al., 2016). In a vanilla VAE, one generative factor changes in the direction of element in a non-standard basis as shown in Fig. 1. With this result, if the latent space can be decomposed with basis $\mathcal{B} = \{\mathbf{b_1}, \cdots, \mathbf{b_n}\}$ to denote latent variable $z$ as $\Sigma_i c_i \mathbf{b_i}$, the single generative factor is associated with the single coefficient. Therefore, disentangled representation learning is achieved by the following two constrains:

1. Each coefficient is subject to change in single generative factor, and relatively invariant to the changes in other factors.

2. The generative model is trained to make the basis of the latent space as a standard basis $\mathcal{B}_0 = \{\mathbf{e_1}, \cdots, \mathbf{e_n}\}$.

In this paper, we focus on the first constraint to formulate disentangled representation without the second constraint. The rest of this paper is organized as follows. In Section 2, we introduce the research for learning disentangled representation. The work we have done in this paper and the proposed model are presented in Section 3 and the evaluation is discussed in Section 4. Section 5 presents a summary and some future works.

## 2 RELATED WORKS

Many studies have been conducted to learn a data representation. It is used on various applications from feature extraction to dimension reduction. Approaches are divided into two categories: conventional methods and deep learning models. Principal component analysis (PCA) or independent component analysis (ICA) are well-known methods to extract features and reduce the size of data (Smith, 2002; Hyvärinen & Oja, 2000). Dictionary learning develops a set (dictionary) of representative elements from the data such that each datum can be expressed as a weighted sum of the atoms. The elements and weights can be found by minimizing the error with L1 regularization on the weights to enable sparsity (Mairal et al., 2009; Lee et al., 2007; Aharon et al., 2006). They adopted the methods such as basis on linear algebra that defines the materials and mixes them appropriately to represent the data. In another approach, Kingma and Welling proposed auto-encoding variational Bayes to approximate the posterior distribution (Kingma & Welling, 2013). Radford et al. showed that the walking in the latent space resulted in semantic changes (Radford et al., 2015). Oord et al. proposed a vector-quantized VAE to learn a discrete latent representation (van den Oord & Vinyals, 2017). It is not disentangled, but somewhat with general representation to prevent posterior collapse (i.e., violation of the first constraint). Chen et al. presented InfoGAN that learned interpretable rep-

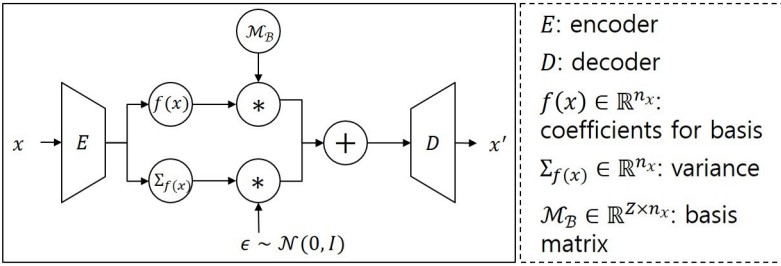

Figure 2: The architecture of the proposed model, BasisVAE.

resentation by using mutual information (Chen et al., 2016). Higgins et al. introduced an adjustable hyperparameter $\beta$ that balanced latent channel capacity and independence constraints with reconstruction accuracy (betaVAE) (Higgins et al., 2017). Dupont improved betaVAE by using a joint distribution of continuous and discrete latent variables (Dupont, 2018). Deep learning frameworks showed promise in disentangling factors of variation, but there was a degrade in the quality of the generated data due to the trade-off. In this paper, we propose a method to learn disentangled representation while maintaining the quality of the generated data by learning materials and weights for data representation like dictionary learning and disentangling factors like deep learning approach.

## 3 THE PROPOSED METHOD

The architecture of a proposed model that constructs disentangled representation (i.e., the association of a single basis element with a single generative factor) with a coefficient of basis element rather than a latent unit is shown in Fig. 2. Unlike the conventional VAE that outputs the mean and variance of the latent space expressed as a normal distribution, the encoder of BasisVAE outputs the coefficient $f(\mathbf{x}) = \mathbf{c}$ associated with elements of the basis B. The latent variable z is sampled from the Gaussian distribution $\mathcal{N}(\mathcal{M}_\mathcal{B} \cdot f(\mathbf{x}), \Sigma_{f(\mathbf{x})})$, where operator $\cdot$ means matrix multiplication, $\Sigma_{f(\mathbf{x})}$ is a variance computed by encoder, and $\mathcal{M}_\mathcal{B} = [\mathbf{b_1}|\cdots|\mathbf{b_n}]$ is a matrix form of bases. The theoretical background, loss function, and algorithms of the proposed model are discussed in detail in the following sections.

### 3.1 LATENT SPACE DECOMPOSITION

For the first constraint mentioned in the introduction, it is proved in Theorem 1 that the latent space can be decomposed as a set of single basis elements that are subject to a single generative factor. It is enough to show that the latent variable z in the equation (2) can be decomposed into latent variables $\mathbf{z_1}, \cdots, \mathbf{z_n}$, called latent basis, associated with a single generative factor, not into latent units, and the evidence lower bound (ELBO) is maintained. Let $n_x$ be the number of features that data x has and $\mathbf{z_1}, \cdots, \mathbf{z_{n_x}}$ be the corresponding independent latent variables. **Theorem 1.** Let the latent variable $\mathbf{z}$ in ELBO be decomposed into independent latent variables $\mathbf{z_1}, \cdots, \mathbf{z_{n_x}}$ associated with a single generative factor such that $p(\mathbf{z}) = \Pi_i p(\mathbf{z_i})$, then the ELBO with respect to $\mathbf{z}$ is equal to the average of values of the ELBO with respect to $\mathbf{z_i}$. The $q_\phi(z|x)$ which the expectation value in equation (1) with respect to should be modified as the form of $q_\phi(z_i|x)_i$. We prove Lemma 1 in order to prove Theorem 1. **Lemma 1.** If $z_1, \cdots, z_n$ are independent and $L$ is a linear operator, $\mathbb{E}_{z_1, \cdots, z_n}[L(z_1, \cdots z_n)] = \Sigma_i L(E_{z_i}[z_i])$ where $a_i$ is a coefficient of $z_i$ in $L$. *Proof.* We just show it in the case of $n = 2$.

$$\mathbb{E}_{z_1, z_2}[L(z_1, z_2)]$$
$$= \int_{z_1} \int_{z_2} p(z_1, z_2)(a_1 z_1 + a_2 z_2) dz_2 dz_1$$
$$= \int_{z_1} p(z_1) a_1 z_1 dz_1 + \int_{z_2} p(z_2) a_2 z_2 dz_2 \tag{3}$$
$$= a_1 \mathbb{E}_{z_1}(z_1) + a_2 \mathbb{E}_{z_2}(z_2)$$
$$= L(\mathbb{E}_{z_1}[z_1], \mathbb{E}_{z_2}[z_2]) \qquad \square$$

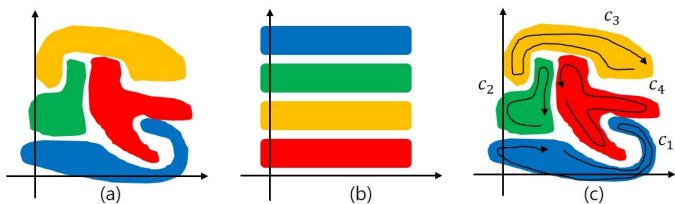

Figure 3: The visualization of (a) the general latent space, (b) the disentangled latent space, and (c) latent space of the proposed model with two coordinates.

*Proof of Theorem 1.* Since the latent variable $z$ can be decomposed as independent latent variable $\mathbf{z_1}, \cdots, \mathbf{z}_{n_x}$, equation (4) is derived from equation (2).

$$\log p_\theta(x) \geq -\mathcal{D}_{KL}[q_\phi(z_1, \cdots, z_{n_x}|x)\|p_\theta(z_1, \cdots, z_{n_x})] \tag{4}$$

As $z_1, \cdots, z_{n_x}$ are independent,

$$\log p_\theta(x) \geq -\mathcal{D}_{KL}[q_\phi(z_1|z_2, \cdots, z_{n_x}, x) \cdots q_\phi(z_{n_x}|x)\|p_\theta(z_1) \cdots p_\theta(z_{n_x})] \tag{5}$$
$$+ \mathbb{E}_{q_\phi(z_1|z_2, \cdots, z_{n_x}, x) \cdots q_\phi(z_{n_x}|x)}[\log p_\theta(x|z_1, \cdots, z_{n_x})]$$

$$\log p_\theta(x) \geq -\mathcal{D}_{KL}[\Pi_i^{n_x} q_\phi(z_i|x)\|\Pi_i^{n_x} p_\theta(z_i)] \tag{6}$$
$$+ \mathbb{E}_{\Pi_i^{n_x} q_\phi(z_i|x)}[\log[\Pi_i^{n_x} p_\theta(x|z_i)/p^{n_x-1}(x)]]$$

$$\log p_\theta(x) \geq -\mathcal{D}_{KL}[\Pi_i^{n_x} q_\phi(z_i|x)\|\Pi_i^{n_x} p_\theta(z_i)] \tag{7}$$
$$+ \mathbb{E}_{\Pi_i^{n_x} q_\phi(z_i|x)}[\log \Pi_i^{n_x} p_\theta(x|z_i)] - (n_x - 1)p(x)$$

By Lemma 1, we can separate the expectation as follows:

$$\log p_\theta(x) \geq \frac{1}{n_x} \Sigma_i^{n_x}[\mathbb{E}_{q_\phi(z_i|x)}[\log p_\theta(x|z_i)] - \mathcal{D}_{KL}[q_\phi(z_i|x)\|p_\theta(z_i)]]\square \tag{8}$$

As a result of equation (8), we can say that the first term of RHS is the reconstruction error, and the second term associates the latent space with the data which has $i$-feature represented as $\mathbf{z_i}$. In the next section, BasisVAE is proposed to maximize the lower bound shown in equation (8), with $\mathbf{z_1}, \cdots, \mathbf{z_{n_x}}$ becoming independent. The proof on the case on $n >= 3$ in Lemma 1 and the derivations from equation (5) to (6) are more discussed in Appendix C.

### 3.2 BASISVAE

We set $n_x$ as the number of features existing in the set $\mathcal{X}$ of data and latent variable $\mathbf{z}$ as linear combination of $\mathbf{z_1}, \cdots, \mathbf{z_{n_x}}$. By the assumption, $\mathbf{z_1}, \cdots, \mathbf{z_{n_x}}$ are independent, and for any $\mathbf{z}$, $\mathbf{z} = \Sigma_i c_i \mathbf{z_i}$ so that the set $\mathcal{B} = \{\mathbf{z_1}, \cdots, \mathbf{z_{n_x}}\}$ is the basis of the latent space. For the sake of convenience, let the elements of $\mathcal{B}$ be denoted as $\mathbf{b_1}, \cdots, \mathbf{b_{n_x}}$. The output of the encoder is coefficients $c_1, \cdots, c_{n_x}$ because, otherwise, the model is not different with vanilla VAE and cannot achieve the disentangled representation. The goal of the previous research is to change the latent space from (a) to (b) in Fig. 3, but the proposed method changes from (a) to (c). It maintains the area responsible for a single generative factor but achieves disentangled representation using coefficient $c_i$. In this method, the model can learn a disentangled representation with coefficients (constraint 1). Besides, it does not have to define the basis of latent space as standard basis (without constraint 2). The direction of the latent basis $\mathbf{b_i}$ is not limited to two (the latent unit becomes larger or smaller), but is set in all directions in $\mathbb{R}^Z$, thus representing the information in various ways.[1] We train the encoder so that $c_i = 1$ and $c_j = 0$ if the input data has $i$-feature and no $j$-feature. The latent variable $\mathbf{z}$ is sampled from the normal distribution $\mathcal{N}(\mathcal{M}_\mathcal{B} \cdot f(\mathbf{x}), \Sigma_{f(\mathbf{x})})$ having the linear combination $\Sigma_i c_i \mathbf{z_i}$ as mean, and $\Sigma_{f(\mathbf{x})}$ as variance, where $\mathcal{M}_\mathcal{B} = [\mathbf{b_1}|\cdots|\mathbf{b_n}]$ and $f(\mathbf{x}) = (c_1, \cdots, c_n)$. The decoder is trained to reconstruct the data $\mathbf{x}$ with $\mathbf{z}$. Algorithm 1 describes the process of defining the latent space through the encoder and reconstructing the data through the decoder. Three losses are defined to train the

---

[1]In $\mathbb{R}^n$, as $n$ increases, the number of direction of $z \in \mathbb{R}^Z$ becomes $2^n$.

---

**Algorithm 1** The process to define the latent space and reconstruct the data

---

1: **Input:** Data $\{x_i\}_{i=1}^N$, encoder $q_\phi$, decoder $p_\theta$, and basis matrix $\mathcal{M}_\mathcal{B}$
2: **Output:** trained encoder $q_\phi$, trained decoder $p_\theta$, and trained basis matrix $\mathcal{M}_\mathcal{B}$
3: Initialize $q_\phi, p_\theta$
4: **for** epochs **do**
5:     **for** batches **do**
6:         Sample $\mathbf{x}$ from the dataset
7:         $c \leftarrow q_\phi(x)$
8:         Sample $z$ from $\mathcal{N}(\mathcal{M}_\mathcal{B} \cdot c^T, \Sigma_{f(x)})$
9:         $\hat{x} \leftarrow p_\theta(z)$
10:        Update $q_\phi, p_\theta, \mathcal{M}_\mathcal{B}$ with equation (12)
11:     **end for**
12: **end for**
13: **return** $q_\phi, p_\theta, \mathcal{M}_\mathcal{B}$

---

latent space in the proposed process: 1) reconstruction loss $\mathcal{L}_{recon}$, 2) inference loss $\mathcal{L}_{KL}$, and 3) basis loss $\mathcal{L}_\mathcal{B}$ as follows.

$$\mathcal{L}_{recon} = l(x, G(\mathcal{C}(z|x))), \mathcal{C}(z|x) \sim \mathcal{N}(\mathcal{M}_\mathcal{B} \cdot f(x), \Sigma_{f(x)}) \tag{9}$$

$$\mathcal{L}_{KL} = \mathcal{D}_{KL}[\mathcal{M}_\mathcal{B} \cdot \mathcal{N}(f(x), \Sigma_{f(x)}) \| p_\theta(z)] \tag{10}$$

$$\mathcal{L}_\mathcal{B} = \|\mathcal{M}_\mathcal{B}^T \mathcal{M}_\mathcal{B} - I\|_2^2 \tag{11}$$

where $l$ is the binary function for measuring the reconstruction error, $f(x)$ is the output of the encoder, and $\mathcal{M}_\mathcal{B} = [b_1|\cdots|b_{n_x}]$ is the basis matrix. Since the elements in $\mathcal{M}_\mathcal{B}$ have to be independent, i.e., $b_i \cdot b_j = 0$ if $i \neq j$, and $b_i \cdot b_i = 1$, $\mathcal{M}_\mathcal{B}^T \mathcal{M}_\mathcal{B}$ should be identity matrix $I$ during training. The total loss of the proposed model is as follows.

$$\mathcal{L} = \alpha \mathcal{L}_{recon} + \beta \mathcal{L}_{KL} + \gamma \mathcal{L}_\mathcal{B} \tag{12}$$

where $\alpha, \beta$, and $\gamma$ are the hyperparameters for balancing between the losses.

## 4 EXPERIMENTS

### 4.1 DATASET AND EXPERIMENTAL SETTINGS

To verify the performance of the proposed model, we use the MNIST, 3DFaces and CelebA datasets (LeCun et al., 1998; Liu et al., 2015; Paysan et al., 2009). The CelebA is a dataset with large-scale face attributes. We crop the initial $178\times218$ size to $138\times138$ and resize them as $128\times128$. There are total 202,599 face images and we use 162,769 images as training data and the rest as test data. The pixel values are normalized between 0 and 1. The weights of the model are initialized with the method proposed by Glorot and Bengio (Glorot & Bengio, 2010). The encoder consists of eight convolutional layers whose filter size is $5\times5$ with LeakyReLU activation function followed by dropout and batch normalization layer (Maas et al., 2013; Srivastava et al., 2014; Ioffe & Szegedy, 2015). The decoder is composed of four convolutional layers and 4 deconvolutional layers with ReLU activation function followed by several layers like encoder (Nair & Hinton, 2010). $\alpha, \beta$, and $\gamma$ are set as 0.0004, 1, and 0.1, respectively. The binary function for measuring the reconstruction error is set as Bojanowski et al. did (Bojanowski et al., 2017). BasisVAE is trained for 100 epochs with 100 batch size. The optimizer used to train the model is Adam proposed by Kingma and Ba (Kingma & Ba, 2014).

### 4.2 GENERATED IMAGES

To verify the performance of the proposed model, the performance of BasisVAE is compared with the performance of vanilla VAE, betaVAE, and VQ-VAE (van den Oord & Vinyals, 2017), which have the same structure, but different output of encoder to the proposed model, in three aspects: Reconstruction, random generation, and disentanglement.

Orig.  Recon. 
(a)

Orig.  Recon. 
(b)

Figure 4: (Top) The original images and (bottom) the reconstructed images on (a) MNIST and (b) CelebA datasets. Appendix A shows more generated images.

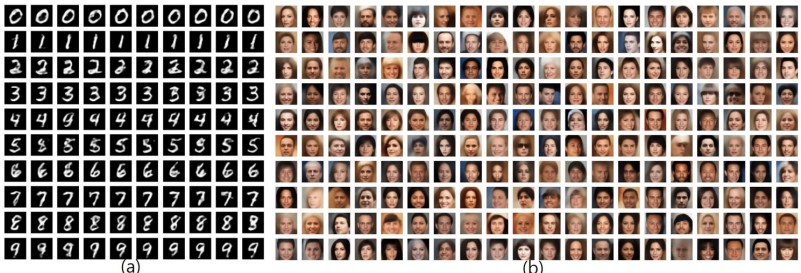

(a) (b)

Figure 5: The generated images of (a) MNIST and (b) CelebA. In the MNIST dataset, the generated data is organized in each row by class.

### 4.2.1 RECONSTRUCTION

We evaluate the reconstruction performance of BasisVAE with MNIST and CelebA datasets. Fig. 4 shows the reconstructed images for the original images. In Table 1, we show the structural similarity (SSIM) and peak signal-to-noise ratio (PSNR) values together with the comparison model for the quantitative evaluation of the performance. The experiment is repeated 10 times to compute the SSIM and PSNR values between the actual images and the generated images by the model trained on CelebA dataset. The results of the t-test show that the performance of the BasisVAE is superior to that of the other models statistically.

### 4.2.2 RANDOM GENERATION

The generated data by BasisVAE learned with MNIST and CelebA are illustrated in Fig. 5. Frechet inception distance is used to evaluate the quality of the generated images (Heusel et al., 2017) as shown in Table 2. The p-value obtained from the t-test was less than 0.05, indicating a statistically significant difference in performance.

### 4.2.3 GENERATION FROM BASIS

We conduct an experiment to verify that the basis learned through BasisVAE has actually influenced the construction of the disentangled representation. BasisVAE generates the images with basis $b_i$ by setting the coefficients as $c_i = 1$ and $c_j$ for $i \neq j$. The feature corresponding to each basis $b_i$ is shown on the Figs. 6 and 7. We also use a 3DFaces dataset as well as CelebA dataset to identify the

Table 1: The results of evaluating the reconstruction performance with SSIM and PSNR.

|      |           | VAE                   | $\beta$ VAE           | VQ-VAE                | BasisVAE              |
|------|-----------|-----------------------|-----------------------|-----------------------|-----------------------|
| SSIM | Average   | 0.7071                | 0.6142                | 0.7564                | **0.7965**            |
|      | Std. dev. | $6.0\times10^{-6}$    | $6.9\times10^{-6}$    | $8.6\times10^{-6}$    | $4.6\times10^{-6}$    |
|      | p-value   | $2.4\times10^{-25}$   | $1.3\times10^{-30}$   | $2.6\times10^{-18}$   | -                     |
| PSNR | Average   | 64.989                | 61.512                | 66.38                 | **67.882**            |
|      | Std. dev. | 0.004                 | 0.004                 | 0.001                 | 0.004                 |
|      | p-value   | $2.1\times10^{2}6$    | $1.1\times10^{-32}$   | $7.8\times10^{-23}$   | -                     |

Table 2: Comparison of image generation quality by FID score.

|  | VAE | $\beta$VAE | VQ-VAE | BasisVAE |
|---|---|---|---|---|
| Average | 112.883 | 168.239 | 84.59 | **78.449** |
| Std. dev. | 1.309 | 1.964 | 7.513 | 2.877 |
| p-value | $1.49\times10^{-21}$ | $1.85\times10^{-28}$ | $5.31\times10^{-6}$ | - |

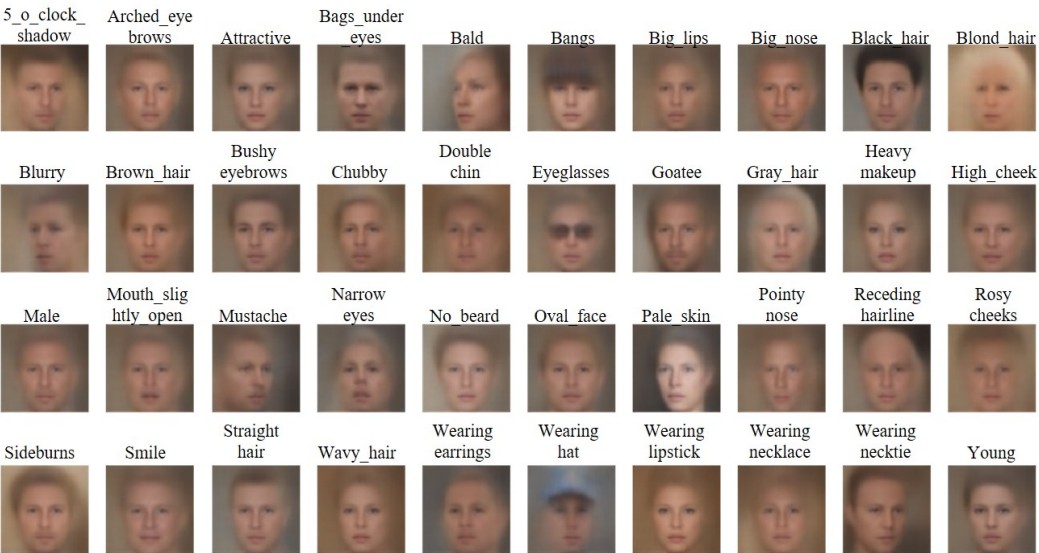

Figure 6: The images generated from a single basis. The corresponding feature is shown above the image.

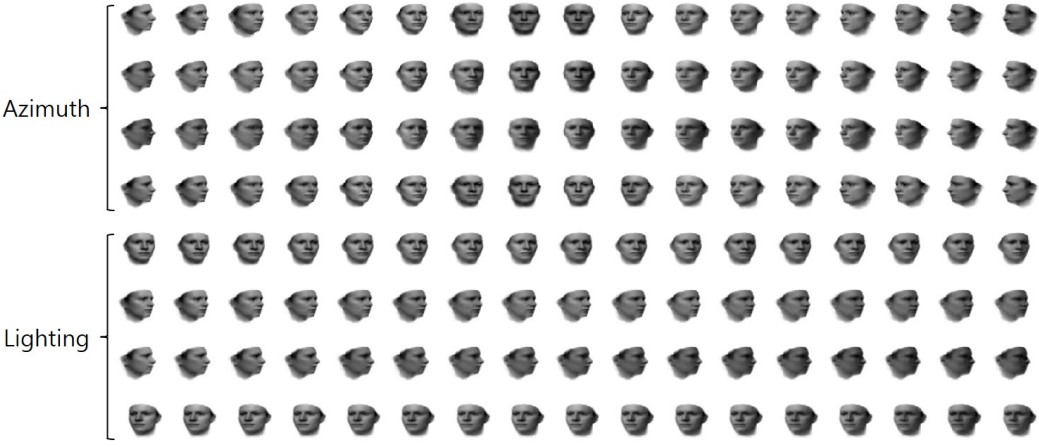

Figure 7: The images generated from a single basis. The value of the coefficient is linearly changed along the row. The corresponding characteristics are shown in the left of the images.

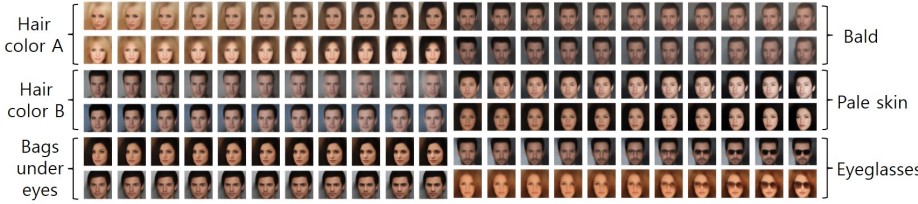

Figure 8: The value of the coefficient is linearly changed along the row. The corresponding characteristics are shown in the side of the images. Appendix A shows more generated images.

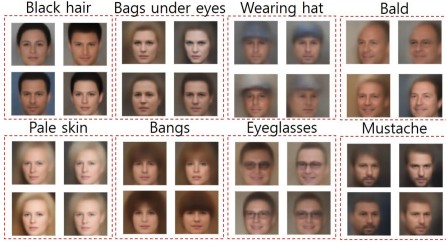

Figure 9: Randomly generated images with the coefficient $c_i$ for $\mathbf{b_i}$ fixed as 1. According to the basis element, the images reflecting the corresponding feature are generated. Appendix A shows more generated images.

characteristic change with coefficient size. As shown in Figs. 8 and 9, we can see the basis element corresponding to azimuth and lighting in 3DFaces dataset and to hair color, bags under eyes, bald, etc. in CelebA dataset. To show that single basis element is subject to a single generative factor, we randomly generate an image with the coefficient $c_i$ for $\mathbf{b_i}$ fixed as 1, as shown in Fig. 9. To quantitatively evaluate the disentangled representation, logistic regression is trained to classify the features by inputting the output of the encoder into itself. The more disentangled the latent space is, the higher accuracy the model achieves. We train about 40 binary classifiers for 40 classes of CelebA dataset, and the average accuracy is shown in Table 3. Appendix A shows the more generated images and Appendix C shows the distribution of coefficients $c_i$ that model learns with examples.

## 5 CONCLUSION

In this paper, we have formulated the disentangled representation learning with two constraints. By proving the Theorem, it is shown that the latent space can be decomposed as independent latent variables associated with single generative factor. We have shown that the proposed BasisVAE constructs disentangled representation without the second constraint by constructing the basis of the latent space. Furthermore, BasisVAE outperforms the vanilla VAE and $\beta$VAE in both performance and disentanglement. Since our method can be applied to other VAEs by changing the output of the encoder as coefficients for basis element and adding loss $\mathcal{L_B}$, we will verify the versatility and validity by applying it to other models. The performance of the proposed model will be evaluated with other well-known benchmark datasets such as CIFAR10, 3DFaces, and ImageNet. In addition, we will achieve the higher quality of the generated data and interpretability of the latent space by constructing disentangled latent space in generative adversarial network.

Table 3: Results of classification using the output of the encoder. The logistic regression model for each class is trained to classify the one class. Appendix B shows more details in the numerical results for each attributes.

|  | VAE | $\beta$VAE | VQ-VAE | BasisVAE |
|---|---|---|---|---|
| Average | 0.8190 | 0.8444 | 0.8225 | **0.8982** |
| Std. dev. | 0.015 | 0.009 | 0.014 | 0.005 |
| p-value | 0.001 | 0.004 | $4.1 \times 10^{-4}$ | - |

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

# A APPENDIX A: MORE IMAGES GENERATED

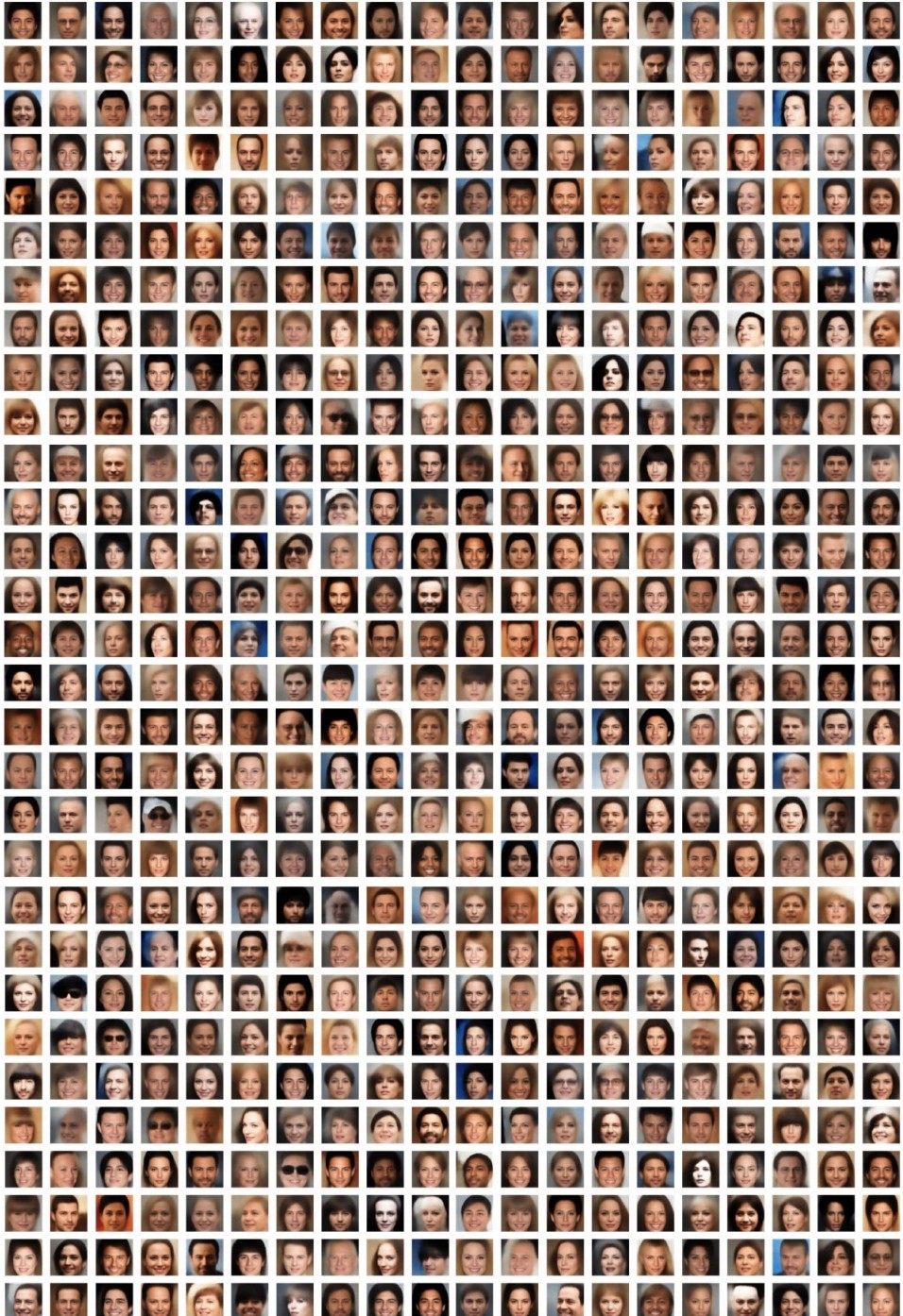

Figure 10: The generated CelebA images. Image blur is less than conventional VAE.

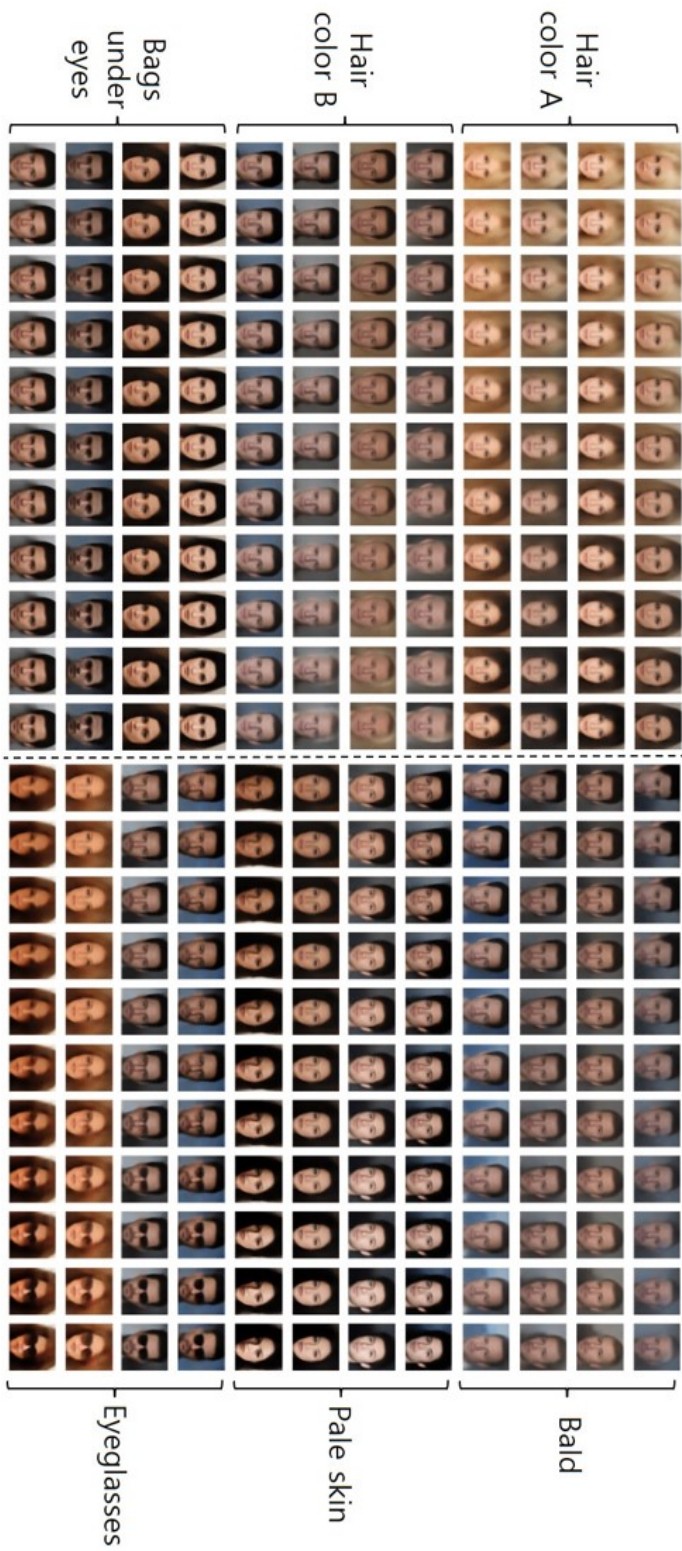

Figure 11: The value of the coefficient is linearly changed along the row. The corresponding characteristics are shown in the side of the images.

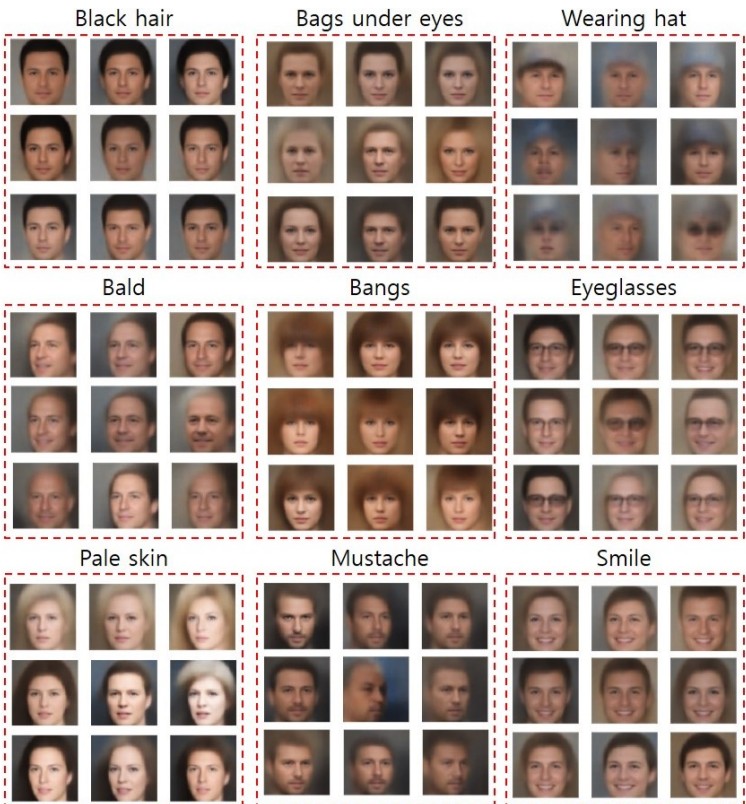

Figure 12: Randomly generated images with the coefficient $c_i$ for $\mathbf{b_i}$ fixed as 1. According to the basis element, images reflecting the corresponding feature are generated.

Table 4: Results of classification using the output of the encoder. The logistic regression model for each class is trained to classify the one class.

| Class | VAE | $beta$VAE | VQ-VAE | BasisVAE |
|---|---|---|---|---|
| 5_o_clock_shadow | 0.8820 | 0.8816 | 0.8820 | **0.9220** |
| Arched_eyebrows | 0.7414 | 0.7450 | 0.7423 | **0.8231** |
| Attractive | 0.6579 | 0.6945 | 0.6468 | **0.7842** |
| Bags_under_eyes | 0.7926 | 0.7920 | 0.7926 | **0.8243** |
| Bald | 0.9798 | 0.9798 | 0.9792 | **0.9808** |
| Bangs | 0.8532 | 0.8890 | 0.8588 | **0.9376** |
| Big_lips | **0.8468** | 0.8466 | **0.8468** | 0.8419 |
| Big_nose | 0.7514 | 0.7568 | 0.7514 | **0.8115** |
| Black_hair | 0.7899 | 0.8335 | 0.7876 | **0.8773** |
| Blond_hair | 0.8470 | 0.8983 | 0.8591 | **0.9377** |
| Blurry | **0.9526** | **0.9526** | **0.9526** | 0.9526 |
| Brown_hair | 0.7586 | 0.7614 | 0.7586 | **0.8304** |
| Bushy_eyebrows | 0.8575 | 0.8603 | 0.8575 | **0.9105** |
| Chubby | 0.9389 | 0.9389 | 0.9389 | **0.9440** |
| Double_chin | 0.9510 | 0.9510 | 0.9510 | **0.9578** |
| Eyeglasses | 0.9305 | 0.9312 | 0.9304 | **0.9837** |
| Goatee | 0.9265 | 0.9265 | 0.92646 | **0.9542** |
| Gray_hair | 0.9513 | 0.9517 | 0.95131 | **0.9702** |
| Heavy_makeup | 0.6430 | 0.7586 | 0.6796 | **0.9037** |
| High_cheeknones | 0.5672 | 0.7116 | 0.5907 | **0.8596** |
| Male | 0.6522 | 0.7720 | 0.6689 | **0.9780** |
| Mouth_slightly_open | 0.5480 | 0.6752 | 0.5542 | **0.9262** |
| Mustache | 0.9496 | 0.9496 | 0.9496 | **0.9519** |
| Narrow_eyes | 0.9250 | **0.9251** | 0.9250 | 0.9222 |
| No_beard | 0.8225 | 0.8288 | 0.8225 | **0.9347** |
| Oval_face | 0.7196 | 0.7189 | 0.7196 | **0.7393** |
| Pale_skin | 0.9569 | 0.9594 | 0.9569 | **0.9625** |
| Pointy_nose | 0.7151 | 0.7151 | 0.7151 | **0.7459** |
| Receding_hairline | 0.9281 | 0.9277 | 0.9281 | **0.9339** |
| Rosy_cheeks | 0.9317 | 0.9316 | 0.9317 | **0.9431** |
| Sideburns | 0.9313 | 0.9314 | 0.9313 | **0.9565** |
| Smiling | 0.5730 | 0.7330 | 0.5957 | **0.9165** |
| Straight_hair | 0.7942 | 0.7947 | 0.7942 | **0.7989** |
| Wavy_hair | 0.7262 | 0.7516 | 0.7288 | **0.7917** |
| Wearing_earrings | 0.8092 | 0.8103 | 0.8092 | **0.8597** |
| Wearing_hat | 0.9528 | 0.9617 | 0.9528 | **0.9798** |
| Wearing_lipstick | 0.6542 | 0.7668 | 0.67945 | **0.9135** |
| Wearing_necklace | **0.8794** | **0.8794** | **0.8794** | 0.8794 |
| Wearing_necktie | 0.9273 | 0.9271 | 0.9273 | **0.9277** |
| Young | 0.7465 | 0.7541 | 0.7464 | **0.8588** |

# B  APPENDIX B: DETAILED RESULTS TO MEASURE DISENTANGLEMENT

The more detailed classification accuracy, summarized in Table 3 on average, are shown in Table 4 by generative factors.

# C  APPENDIX C: MORE DETAILS IN THE PROOF OF LEMMA 1 AND THEOREM 1

In the Section 3.1, we show the proof of Lemma 1 in the case of $n = 2$. For more generality, it is necessary to be proved that the Lemma 1 holds in the case of $n >= 3$. In that case, Let $z_2 = (z_3, z_4, \ldots, z_n)$, repeat the same verification process, then we can separate the $z_3$ from the $(z_4, \cdots, z_n)$. If we process it recursively, the Lemma 1 in the case of $n >= 3$ holds too.

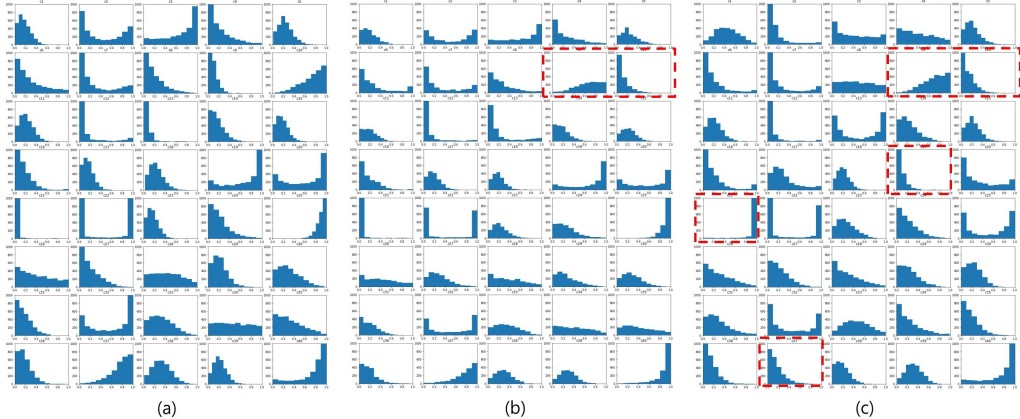

(a)                         (b)                        (c)

Figure 13: The distribution of $c_i(1 <= i <= 40)$ that model learns for each data with (a) "*female with blond hair*", (b) "*female with black hair*", and (c) "*male with black hair*". We have red boxes for coefficients where the difference between distributions in (a) and distributions in (b) and c is apparent.

As in the proof of Lemma 1, it is enough to prove the Theorem 1 that Theorem 1 holds when $n = 2$. The flow of the equations (4) to (8) except for the equations (5) to (6) is natural. Equation (6) is derived from the (5) with $p(x \mid z_1, z_2) = p(x \mid z_1)p(x \mid z_2)/p(x)$, which is equivalent to equation (13).

$$\frac{p(x, z_1, z_2)}{p(z_1, z_2)} = \frac{p(x, z_1)p(x, z_2)}{p(x)p(z_1)p(z_2)} \tag{13}$$

Since $z_1, \cdots, z_n$ are independent, it is drived to (14).

$$p(x, z_1, z_2) = \frac{p(x, z_1)p(x, z_2)}{p(x)} \tag{14}$$

The LHS of equation (14) can be modified as (15) to (18)

$$\begin{aligned} p(x, z_1, z_2) &= \frac{p(x, z_1, z_2)}{p(x, z_1)}p(x, z_1) \\ &= p(z_2 \mid x, z_1)p(x, z_1) \\ &= p(z_2 \mid x)p(x, z_1) \\ &= \frac{p(x, z_1)p(x, z_2)}{p(x)} \end{aligned} \tag{15}$$

Since the LHS of equation (14) can be derived to the RHS of equation (15), the equation (13) is correct, resulting in the correctness of Theorem 1.

## D   APPENDIX D: THE DISTRIBUTION OF COEFFICIENTS $c_i$

We show the distribution of $c_i$ that the model learns for each data with "*female with blond hair*", "*female with black hair*", and "*male with black hair*" in the Fig. 13. With comparison of distributions in Fig. 13(a) and (b), coefficients $c_9$ and $c_{10}$ represent "*black hair*" and "*blond hair*", respectively. Furthermore, we can confirm that coefficients $c_{19}$, $c_{21}$, and $c_{37}$ represent the generative factors related to the characteristics on gender, comparing Fig. 13(a) and (c).

