# OpenReview forum: "BasisVAE: Orthogonal Latent Space for Deep Disentangled Representation"
_ICLR.cc/2020/Conference — Reject_

### Official Review · AnonReviewer3 · 2019-10-17
**Official Blind Review #3**

**Rating:** 1

**Review:**

This paper proposes BasisVAE for acquiring a disentangled representation of VAE.
Though the topic is much of interest for this conference, I cannot support its acceptance because the paper leaves many aspects unexplained in the model design.

In particular, the following points need justified and clarified.
1) Theorem 1 is difficult to follow.
The claim of the theorem is unclear.
I suppose it says ELBO can be written as a sum with respect to z_i given p(z)=\prod_i p(z_i), but the statement is not clear enough from the text.
Proof of Lemma 1 is logically incomplete. Discuss the cases n>2.
Derivation of equation (6) from (5) seems erroneous: p(x|z_1, ..., z_n) = \prod_{i=1}^n p(x|z_i) / p^{n-1}(x) does not hold in general even if z_i's are independent p(z_1, ..., z_n)=\prod_{i=1}^n p(z_i).

2) Connection between the objective function and Theorem 1 is unclear.
BasisVAE uses a linear combination of Eqs. (9,10,11) as its objective function.
How Theorem 1 motivates this formulation?

3) Reconstruction error (9).
The text says \ell of Eq. (9) is the binary function and configured as in (Bojanowski et al. 2017).
However, Bojanowski et al. used a weighted l1 error Laplacian Pyramid representation.
Furthermore, the original VAE formulation uses a conditional log-likelihood log p(x|z) for the reconstruction term.
How is binary function \ell related the likelihood?

4) KL regularization term (10).
For computing this term, the output of encoder c=f(x) should be converted into z.
Notation of N(f(x), \Sigma) is confusing.

5) Figure 6 shows diversity in many factors.
Figure 6 is not as impressive for disentangled images since many factors change by varying a single basis.
Is this an expected result?

**Experience Assessment:**

I have read many papers in this area.

**Review Assessment: Checking Correctness Of Derivations And Theory:**

I carefully checked the derivations and theory.

**Review Assessment: Checking Correctness Of Experiments:**

I assessed the sensibility of the experiments.

**Review Assessment: Thoroughness In Paper Reading:**

I read the paper at least twice and used my best judgement in assessing the paper.

---

> ### Author Response · Authors · 2019-11-11
> **Answers to Reviewer #3**
>
> Thank you for your comments. They are very helpful for us to conduct more finished works. According to the reviewer’s comments, we have addressed them as follows.
> 	1. It is enough to show p(x│z_1,z_2 )=(p(x│z_1 )p(x│z_2 ))/(p(x)) for derivation from (5) to (6). We have added it in Appendix C.
> 	2. We derive from Equation 8 that a latent variable z can be decomposed into several independent variables z_i, generating the same data x from them with the encoder, and constructing an ELBO. In the BasisVAE, z_i corresponds to the basis element b_i, and it is adjusted by the coefficient c_i output of the encoder.
> 	3. A binary function is a function that takes two arguments and becomes cross-entropy as in VAE or weighted l1 error Laplacian Pyramid as in Bojanowski et al.
> 	4. Sorry for the typos. N(f(x),\Sigma_f(x)) should be replaced with N(M_B*f(x),\Sigma_f(x)). We have corrected it.
> 	5. Fig. 6 shows the result when only one c is 1 and the others are 0. It is shown that the basis elements have one distinct characteristic and only one characteristic changes in Fig. 8 when changing the strength of the basis element (i.e., c). More examples are shown in Figure 11. These results are seen in MNIST and 3DFace datasets as well as CelebA datasets in Figures 5 and 7. In addition, we also demonstrate the performance by showing the quantitative evaluation of disentanglement in Table 3.

---

> > ### Comment · AnonReviewer3 · 2019-11-13
> > **Point 1. (Issue 1 of Reviewer #2)**
> >
> > Thank you for the response.
> >
> > As in the discussion with Reviewer #2, I am not yet convinced that p(z_1, z_2 | x) factorizes into p(z_1|x)p(z_2|x).
> > The generative model uses decoder $x \sim p_\theta(\cdot | z)$ (line 9 of Algorithm 1) where the decoder network takes all components of $z$ as its input.
> > In this case, the conditional independence $p(z_1|x)p(z_2|x)$ should be carefully justified. (I think the decoder needs some special structure.)
> >
> > Regarding point 3: I interpreted the binary function as a function that returns a binary value {0,1}.
> > Simply calling "negative log likelihood" or "reconstruction function" can avoid the possible confusion.

---

> > > ### Author Response · Authors · 2019-11-14
> > > **Answers to Reviewer #3**
> > >
> > > Thank you for your respond!
> > >
> > > 1.  We can derive $p(z_1, z_2 | x)=p(z_1 |z_2 , x)p(z_2 | x)$ with a conditional probability. In the assumption that the $z_i$ are independent conditioned by $x$, since $p(z_1 | z_2 , x) =p(z_1 |x)$, $p(z_1 , z_2 |x)$ can be factorized into $p(z_1 |x)p(z_2 |x)$. The $z$ in the line 9 of Algorithm 1 is sampled from $N(M_B \cdot c^T,\Sigma_{f(x)})$. In this process, we intended that the encoder outputs coefficient $c_i$ for independent $z_i$, and the decoder generates data by inputting $z$ which is a linear combination of $c_i$ and $z_i$. Therefore, decoder takes $z$ which is a linear combination of all components of $z$.
> > >
> > > 2. Thank you for your comments. To avoid the confusion, we'll correct that word.

---

> > > > ### Comment · AnonReviewer3 · 2019-11-14
> > > > **Encoder does not assure the assumption**
> > > >
> > > > The conditional independence $p(z_1, z_2|x)=p(z_1|x)p(z_2|x)$ should be derived from the configurations of the prior $p(z)$ and decoder $p_\theta(x|z)$.
> > > > How you sample $z$ from the encoder (or your variational distribution) is irrelevant of the conditional independence of the *true* posterior.
> > > >
> > > > This is why I indicate that the decoder (or maybe the prior) needs a special structure.

---

> > > > > ### Author Response · Authors · 2019-11-14
> > > > > **Answers to Reviewer #3**
> > > > >
> > > > > We considered conditional independence in this paper in relation to disentanglement. That is, with $x$ and corresponding $z=\Sigma c_i z_i$, the feature changed by $z_1$ and the feature changed by $z_2$ are not related to each other. Therefore, when conditional by $x$, the probability that the property represented by $z_1$ and $z_2$ $p(z_1 , z_2|x)$ is expressed as the probability that the property represented by z_2 multiplied by the probability that the property represented by z_1 is expressed $p(z_1 |x)p(z_2 |x)$.
> > > > >
> > > > > Therefore, the decoder needs to get $z_i$ from one of the encoder's outputs, $c_i$, and generate $x$ from it, and then this process is proceeded and the loss is calculated for all $i$. But, for the convenience, decoder generates $x$ from the linear combination $z=\Sigma c_i z_i$.

---

### Official Review · AnonReviewer1 · 2019-10-23
**Official Blind Review #1**

**Rating:** 1

**Review:**

[updated rating due to supervision of $c_i$, which was not made clear enough and would require other baseline models]

This paper proposes a modification of the usual parameterization of the encoder in VAEs, to more allow representing an embedding $z$ through an explicit basis $M_B$, which will be pushed to be orthogonal (and hence could correspond to a fully factorised disentangled representation). It is however possible for different samples $x$ to use different dimensions in the basis if that is beneficial (i.e. x is mapped to $z = f(x) \cdot M_B$, where f(x) = (c_1, ... , c_n) which sums to 1.). This stretches the usual definition of what a “disentangled representation” means, as this disentanglement is usually assumed to be globally consistent, but this is a fair extension.
They show that this formulation can be expressed as a different ELBO which can be maximized as for usual VAEs.

I found this paper interesting, but I have one clarification that may modify my assessment quite strongly (hence I am tentatively putting it on the accept side). Some implementation details seem missing as well. Otherwise the presentation is fair, there are several results on different datasets which demonstrate the model's behaviour appropriately.

1.	The main question I have, which may be rather trivial, is “are the c_i supervised in any way?”. When I first read the paper, and looking at the losses in equations 9-11, I thought that this wasn’t the case (also considering this paper is about unsupervised representation learning), but some sentences and figures make this quite unclear:
	a.	In Section 3.2, you say “We train the encoder so that c_i = 1 and c_j = 0 if the input data has i-feature and no j-feature”. Do you?
	b.	How are the features in Figure 6 attached to each b_i? I.e. how was “5_o_clock_shadow” attached to that particular image at the top-left?
	If the c_i are supervised, this paper is about a completely different type of generative modeling than what it compares against (it would be more comparable to VQ-VAE or other nearest-neighbor conditional density models).
2.	There is not enough details about the architecture, hyperparameter and baselines in the current version of the paper.
	a.	What n_x (i.e. dimensionality of the basis) do you use? How does this affect the results?
	b.	How exactly are f(x), \Sigma_f(x) parametrized? They mention the architecture of the “encoder” in Section 4.1, but this could be much clearer.
	c.	How do you train M_B? I assume they are just a fixed set of embeddings that are back-propagated through?
	d.	What are the details about the architecture of the baselines, and their hyperparameters? E.g. what is the beta you used for Beta-VAE?
3.	The reconstructions seem only partially related to their target inputs (e.g. see Figure 4). This seems to indicate that instead of really reconstructing x, the model chooses to reconstruct “a close-by related \tilde{x}”, or even perhaps a b_i. This would make it behave closer to VQ-VAE, which explicitly does that. How related are reconstructions/samples to the b_i?
4.	Could you show the distribution of c_i that the model learns, and how much they vary for several example images?  How “peaky” is this distribution for a given image (this feeds into to the previous question as well)? The promise of the proposed model is that different images pick and choose different combinations of b_i, which hopefully one should see reflected in the distributions of c_i per sample, across clusters, or across the whole dataset.
5.	What happens when L_B is removed? I.e. what is the effect of removing the constraint on M_B being a basis, and instead allow it to be anything? This seems to make it closer to a continuous approximation to VQ-VAE?
6.	Is Equation 10 correct? Should the KL use N(f(x) \cdot M_B, \Sigma_f(x)), as in equation 9 above?
7.	Similarly, in Section 4.2.3, did you mean “c_i = 1 and c_j = 0 for i != j”?

If the model happens to be fully unsupervised, I think that these results are quite interesting, and provide a good modification to the usual VAE framework, I find that having access to the M_B basis explicitly could be very valuable.

There is still an interesting philosophical discussion to be had about when one would like to obtain a “global basis” for the latent space (i.e. Figure 3 (b)), or when one would prefer more local ones. I can see clear advantages for a non-local basis, in terms of generalisation and compositionality, which your choice (i.e. Figure 3 (c) ) would prohibit.

References:
[1] VQ-VAE: Aaron van den Oord, Oriol Vinyals, Koray Kavukcuoglu, “Neural Discrete Representation Learning”, https://arxiv.org/abs/1711.00937

**Experience Assessment:**

I have published in this field for several years.

**Review Assessment: Checking Correctness Of Derivations And Theory:**

I assessed the sensibility of the derivations and theory.

**Review Assessment: Checking Correctness Of Experiments:**

I carefully checked the experiments.

**Review Assessment: Thoroughness In Paper Reading:**

I read the paper thoroughly.

---

> ### Author Response · Authors · 2019-11-11
> **Answers to Reviewer #1**
>
> Thank you for your comments. They are very helpful for us to conduct more finished works. According to the reviewer’s comments, we have addressed them as follows.
> 1.	We conducted the experiments with supervised learning, but we have obtained similar results when repeating all the experiments with unsupervised learning. In response to the reviewer's comment, we have also added a comparison with the VQ-VAE model.
> Figure 6 can be verified according to the relationship between the distribution of coefficient c and the characteristics of the input image.
> 2.	We set n_x to 40 according to our previous work. For larger n_x values, there was no significant difference, but in small cases, more than two generative factors appear on one basis element.
> As shown in Figure 2, f(x)=(c,\sigma), i.e., encoder outputs the coefficient and \sigma simultaneously as in VAE. Besides, the basis matrix B can be trained with equation (11) as in VQ-VAE.
> As mentioned in Section 4.2, The layer structure of the model is almost similar, and sampling z is performed using encoder f(x) and \sigma with no basis compared to the proposed model. In betaVAE, beta is set to 100 times the coefficient of the reconstruction error.
> 3.	Thank you for the good comment. We already quantitatively assessed the reconstruction performance and listed it in Table 1 and confirmed that it showed the best performance. In fact, our model puts forward the theory of decomposing the latent space and built the basis to perform it, and makes the main contribution to the advantages (especially on disentanglement) that can be obtained by constructing the latent variable from the linear combination of the bases.
> 4.	 Thank you for the good comment. We describe in appendix D the results of investigating differences in c_i distributions for "blonde women", "black-haired women" and "black-haired men". We will continue to add the comparisons of distribution for the various samples.
> 5.	By removing L_B, the basis elements are not orthonormal to each other, so the Cartesian coordinate system is not set by default with that kind of basis. Thus, there will be more relationships between the basis elements, and the disentanglement will disappear.
> 6.	Sorry for the typos. N(f(x),\Sigma_f(x)) should be replaced with N(M_B*f(x),\Sigma_f(x)). We have corrected it.
> 7.	To avoid the confusion, we have corrected it. Thank you for your comments.

---

> > ### Comment · AnonReviewer1 · 2019-11-14
> > **Response to authors**
> >
> > Thank you for the thorough rebuttal and for adding the comparison to VQ-VAE, very helpful.
> >
> > 1. Thanks for making this clear. Unfortunately, I actually feel that the fact that you supervise $c_i $ but do not make this clear is rather problematic.
> > All baselines you compare against, and the history of the field of disentanglement learning is addressing the issue of *unsupervised* representation learning. Checking again, I see that you do not mention "unsupervised" or "supervised" anywhere in the paper, so this is more of an assumption on my side, but I feel you should have make that more explicit, or report unsupervised results in the Appendix.
> >
> > It also puts you in competition with a plethora of other works which actually leverage supervision (fully supervised or semi-supervised). For example the Multimodal VAE literature has been doing this for a while [1-7], which are missing from the Related work.
> >
> > Unfortunately, due to this issue, I do not feel comfortable with approving the manuscript in this state, as this would require quite a rewrite and change of baselines.
> >
> > References:
> > [1] For a good minimal extension of the VAE framework to introduce supervision, and their Related work section: https://arxiv.org/abs/1906.01044
> > [2] Siddarth et al 2017, https://arxiv.org/abs/1706.00400
> > [3] DC-IGN: https://arxiv.org/abs/1503.03167
> > [4] JMVAE: https://arxiv.org/abs/1611.01891
> > [5] BiVCCA: https://arxiv.org/abs/1610.03454
> > [6] TELBO: https://arxiv.org/abs/1705.10762
> > [7] MVAE: https://arxiv.org/abs/1802.05335
> > [8] Beta-TCVAE: https://arxiv.org/abs/1802.04942
> >
> >
> > --
> > Other points:
> >
> > 4. Thanks for the added plot. It was particularly informative to indicate specific sets of c_i that switch their behaviour. It would be good to see this for a single image, as currently this makes it hard to know what all the other c_i end up doing. Are they just capturing variability around a canonical c_j (which would be like the prototype / mean vector like in VQ-VAE)?
> >
> > 5. I understood what the term does, I was wondering if you observed this lack of disentanglement happening *in practice*?
> >
> > --
> >
> > As a point towards addressing the issue that the other reviewers have with the ELBO formulation:
> >
> > The proposed ELBO derivation is not helpful in practice, because looking at the loss terms 9-11, none of them assume the fully factorised form shown in equation 8.
> > Instead, BasisVAE simply forces the encoder to have a specific parametrisation (a linear combination of K vectors: z = \sum_i c_i m_i), but this is never strictly enforced (even the basis assumption of $M_B$, which would be required for Equation 8 to be really used, is only a regularisation term...).
> > This choice of parametrisation may be unable to capture some distributions p(x), which is a trade-off to decide for the user.
> >
> > Hence personally I would remove Theorem 1 as it does not help and is confusing at best for people that expect a VAE to capture any distribution p(x).

---

### Official Review · AnonReviewer2 · 2019-10-28
**Official Blind Review #2**

**Rating:** 1

**Review:**

Summary:
This paper claims to achieve disentanglement by encouraging an orthogonal latent space.

Decision: Reject. I found the paper difficult to read and the theoretical claims problematic.

Issue 1: The Theorem
Can the authors explain how they got from Eq 5 to Eq 6? It seems that the authors claim that:
p(x | z1 z2 … zn) = p(x | z1) … p(x | zn) / p(x)**(n - 1)
I have difficulty understanding why this is true. It would suggest that
p(x | a b) = p(x | a) p(x | b) / p(x).
Suppose a and b are fair coin flips and x = a XOR b. Then
p(x=1 | a=1 b=1) = 0
p(x=1 | a=1) = 0.5
p(x=1 | b=1) = 0.5
p(x=1) = 0.5
Can the authors please address this issue?

Even if Equation 8 is somehow correct, can the authors explain why BasisVAE provably maximizes the RHS expression in Eq 8? In particular the object p(x | z_i) is the integral of p(x, z_not_i | z_i) d z_not_i, which is quite non-trivial.

Issue 2: The Model
The notation is a bit confusing, but it looks like the proposed model is basically a standard VAE, but where the last layer of the mean-encoder is an orthogonal matrix. I do not think the authors provided a sufficient justification for how this model relates back to Theorem 1.

Furthermore, it is unclear to me why an orthogonal last-layer is of any significance theoretically. Suppose f is a highly expressive encoder. Let f(x) = M.T g(x) where g is itself a highly expressive neural network. Then M f(x) = g(x), which reduces to training a beta-VAE (if using Eq 12). From a theoretical standpoint, it is difficult to assess what last-layer orthogonality is really contributing.

Issue 3: The Experiments
Experimentally, the main question is whether the authors convincingly demonstrate that BasisVAE achieves better disentanglement (independent of whether BasisVAE is theoretically well-understood).

The only experiment that explicitly compares BasisVAE with previous models is Table 3. What strikes me as curious about the table is the standard deviation results. They are surprisingly small. Did the authors do multiple runs for each model? Furthermore, the classification result is not equivalent to measuring disentanglement. There exists examples of perfectly entangled representation spaces can still achieve perfect performance on the classification task (any rotation applied to the space is enough to break disentanglement if disentanglement is defined as each dimension corresponding to a single factor of variation).

**Experience Assessment:**

I have published one or two papers in this area.

**Review Assessment: Checking Correctness Of Derivations And Theory:**

I carefully checked the derivations and theory.

**Review Assessment: Checking Correctness Of Experiments:**

I carefully checked the experiments.

**Review Assessment: Thoroughness In Paper Reading:**

I read the paper thoroughly.

---

> ### Author Response · Authors · 2019-11-11
> **Answers to Reviewer #2**
>
> Thank you for your comments. They are very helpful for us to conduct more finished works. According to the reviewer’s comments, we have addressed them as follows.
> Issue 1
> 	1. It is enough to show p(x│z_1,z_2 )=(p(x│z_1 )p(x│z_2 ))/(p(x)) for derivation from (5) to (6). We have added it in Appendix C.
> 	2. We derive from Equation 8 that a latent variable z can be decomposed into several independent variables z_i, generating the same data x from them with the encoder, and constructing an ELBO. In the BasisVAE, z_i corresponds to the basis element b_i, and it is adjusted by the coefficient c_i output of the encoder.
>
> Issue 2
> 	1. The output of the encoder is coefficient c_i, which is multiplied by the basis matrix and added to \epsilon * \sigma to produce a latent variable z. We have shown that latent space can be decomposed in Thm 1, which shows that latent variable z can be represented as a linear combination of several basis elements. It can be done with less constrains than conventional disentanglement representation, resulting in more effective method.
> 	2. M satisfying M.T * M = I may have many cases besides identity matrix I. In the case of the conventional disentanglement representation method, M = I is made so that a single latent unit is associated with a single generative factor. However, in the proposed method, a single basis element is associated with a single generative factor, which is free from the second constraint mentioned in Section 1.
>
> Issue 3
> 	1. We have slightly simplified the disentanglement-specific metric used in betaVAE as the performance of the simplest logistic regression (LR) using the coefficient c (or latent variable z) extracted through the encoder. As mentioned by the reviewer, rotation is applied. Nevertheless, the results show that the proposed model has the simplest design of latent space, which makes it easier to distinguish generative factors.
> 	2. Sorry for the confusion. In the first original, average was in %???. We have made the appropriate modifications to avoid the confusion.
>
> According to the comments, we have made up the lack of explanation in main contents and added more stuffs such as the results of VQ-VAE for comparison and the distribution of coefficient c_i at the appendix.

---

> > ### Comment · AnonReviewer2 · 2019-11-11
> > **Response to Issue 1**
> >
> > Thanks for the response. I'd like to address Issue 1 first.
> >
> > I checked your Appendix C and noticed the following claim:
> >
> > $p(z_2 | x, z_1)  = p(z_2 | x)$
> >
> > I don't think this claim is correct in general (see "explaining away" effect in v-structures). Can the authors clarify this step?

---

> > > ### Author Response · Authors · 2019-11-11
> > > **Answers to Reviewer #2**
> > >
> > > Thank you for your respond!
> > >
> > > We can check the equation in the derivation of the naive Bayes classifier [1-3].
> > > The "naive" conditional independence assumptions in the naive Bayes classifier come into play on our derivation: assume that all latent variables in \mathbf{z} are mutually independent, conditional on the data \mathbf{x}. Under this assumption,
> > > p(z_i|z_{i+1},...,z_n,x) = p(z_i|x)
> > >
> > > [1] Ceci, M., Appice, A., & Malerba, D. (2003). Mr-SBC: a multi-relational naive bayes classifier. In European conference on principles of data mining and knowledge discovery, 95-106.
> > > [2] Hilden, J. (1984). Statistical diagnosis based on conditional independence does not require it. Computers in biology and medicine, 14(4), 429-435.
> > > [3] Domingos, P., & Pazzani, M. (1997). On the optimality of the simple Bayesian classifier under zero-one loss. Machine learning, 29(2-3), 103-130.

---

> > > > ### Comment · AnonReviewer2 · 2019-11-11
> > > > **Potential misuse of Naive Bayes assumption**
> > > >
> > > > Assuming that all the latent variables are statistically independent conditional on the data is a very big assumption.
> > > >
> > > > In the standard Naive Bayes setup, this assumption is fundamentally baked into the model class (by virtue of the PGM, every model within the Naive Bayes model class provably satisfies the Naive Bayes assumption: whereby the observed features are assumed to be independent when conditioned on the underlying class).
> > > >
> > > > In contrast, your proof assumes that p(z_{1:k} | x) = prod_i p(z_i | x) within a VAE model class, which is not something that's actually guaranteed by the VAE model class.
> > > >
> > > > I am therefore quite uncomfortable with the analysis in Section 3.1. As of the moment, I am strongly inclined to believe that Theorem 1 is wrong. Please let me know what you think.

---

> > > > > ### Author Response · Authors · 2019-11-12
> > > > > **Answers to Reviewer #2**
> > > > >
> > > > > Thank you for your quick response!
> > > > > I am pleased to be able to conduct this constructive discussion with you.
> > > > >
> > > > > Our latent variables are split into multiple levels z_1, ..., z_n. The joint posterior over all of these is a simple fully factorized Gaussian (e.g. conditioned on x, z_2 is independent of z_1), unlike normalizing flows which are used to make the posterior distribution more flexible.
> > > > >
> > > > > Besides, as in [1], if you look at the equation associated with -L (x, q, p) on page 4, you can see that the same assumption is used when moving from the first expression to the second.
> > > > >
> > > > > [1] Gulrajani, I., Kumar, K., Ahmed, F., Taiga, A. A., Visin, F., Vazquez, D., & Courville, A. (2016). Pixelvae: A latent variable model for natural images. International Conference on Learning Representation.

---

> > > > > > ### Comment · AnonReviewer2 · 2019-11-12
> > > > > > **Response**
> > > > > >
> > > > > > It is true that for hierarchical VAEs whose PGM is $z_1 \to z_2 \to \cdots \to x$, that $p(z_i \mid x, z_j) = p(z_i \mid z_j)$ when $i < j$.
> > > > > >
> > > > > > However, your model, from what I can tell, is not a hierarchical VAE. So I don't quite understand your claim that your "latent variables are split into multiple levels z_1, ..., z_n".
> > > > > >
> > > > > > Furthermore, this is still not the same as your claim that $p(z_i, z_j \mid x) = p(z_i \mid x)p(z_j \mid x)$.
> > > > > >
> > > > > > And regarding the PixelVAE paper, the assumption that $q(z \mid x)$ is factorized is an explicit assumption on the variational inference model, which is not the same object as the true posterior $p(z \mid x)$ of the generative model in the VAE.
> > > > > >
> > > > > > Can you clarify what you meant?

---

> > > > > > > ### Author Response · Authors · 2019-11-12
> > > > > > > **Answers to Reviewer #2**
> > > > > > >
> > > > > > > I'm sorry for using confusion expression. We mean "Multiple levels" that the "multiple disentangled latent vector".
> > > > > > > Basically, the proposed model is related to the disentanglement representation. By definition, in that space, a latent variable covers only one generative factor and does not affect each other [1, 2], which can be interpreted as independence. We put the latent space disentangled in Theorem 1 and each factor at that time is z_1, ..., z_n. (This is evidenced by experiments with only one latent variable changed in many disentanglement representation studies [3, 4].)
> > > > > > >
> > > > > > > [1] Y. Bengio, A. Courville, and P. Vincent. Representation learning: A review and new perspectives.
> > > > > > > IEEE Trans. on Pattern Analysis and Machine Intelligence, 35(8):1798–1828, 2013.
> > > > > > > [2] K. Ridgeway. A survey of inductive biases for factorial representation-learning. arXiv preprint
> > > > > > > arXiv:1612.05299, 2016.
> > > > > > > [3] Higgins, I., Matthey, L., Pal, A., Burgess, C., Glorot, X., Botvinick, M., ... & Lerchner, A. (2017). beta-VAE: Learning Basic Visual Concepts with a Constrained Variational Framework. ICLR, 2(5), 6.
> > > > > > > [4] Chen, X., Duan, Y., Houthooft, R., Schulman, J., Sutskever, I., & Abbeel, P. (2016). Infogan: Interpretable representation learning by information maximizing generative adversarial nets. In Advances in neural information processing systems (pp. 2172-2180).

---

> > > > > > > > ### Comment · AnonReviewer2 · 2019-11-12
> > > > > > > > **Please Rewrite Theorem 1**
> > > > > > > >
> > > > > > > > It looks like we're not on the same page. I believe this confusion is arising from our disagreement over how to read Theorem 1. All I can say with confidence at the moment is that, in the standard VAE setup, the jump from Eq 5 to 6 is wrong.
> > > > > > > >
> > > > > > > > If you wish to convince me otherwise, please restate Theorem 1 and its proof as rigorously as possible.

---

> > > > > > > > > ### Author Response · Authors · 2019-11-12
> > > > > > > > > **Answers to Reviewer #2**
> > > > > > > > >
> > > > > > > > > Theorem 1 is true if z_i are independent conditioned by x.
> > > > > > > > >
> > > > > > > > > We found that in Figure 1, only one feature changes with z in the normal VAE. This is represented by z = c1z1 + c2z2 in a two-dimensional representation, meaning that only one feature is adjusted according to c, and z1 and z2 are disentangled, but not on a standard basis.
> > > > > > > > > We proceed on the assumption that z_i are independent when disentangled. I apologize that this has made you very confused. We will add detailed and in-depth assumptions and content about what you pointed out.
> > > > > > > > >
> > > > > > > > > Thanks again for the good point.

---

> > > > > > > > > > ### Comment · AnonReviewer2 · 2019-11-12
> > > > > > > > > > **What is the significance of Theorem 1?**
> > > > > > > > > >
> > > > > > > > > > Does this mean that "z_i are independent conditioned by x" is an explicit assumption in the premise of Theorem 1? If so, OK, I accept that Theorem 1 is correct.
> > > > > > > > > >
> > > > > > > > > > But what, then, is the significance of Theorem 1? Since the premise for Theorem 1 is violated in a VAE, then Theorem 1 can't be applied to a VAE.

---

> > > > > > > > > > > ### Author Response · Authors · 2019-11-12
> > > > > > > > > > > **Answers to Reviewer #2**
> > > > > > > > > > >
> > > > > > > > > > > Theorem 1 shows that the existing ELBO can be separated into independent z_i's.
> > > > > > > > > > > Based on these observations, we set the output of the encoder to coefficient c_i for independent z_i instead of one integrated z, as in normal VAE, even though this actually violated to VAE.
> > > > > > > > > > > By setting the loss as equations (9) ~ (11), we have trained the data representation to separate the z_i from each other (ie, to satisfy the disentanglement).

---

> > > > > > > > > > > > ### Comment · AnonReviewer2 · 2019-11-12
> > > > > > > > > > > > **Potentially false premise**
> > > > > > > > > > > >
> > > > > > > > > > > > The statement "Theorem 1 shows that the existing ELBO can be separated into independent z_i's." is only true if we believe Theorem 1's premise that "z_i are independent conditioned by x" is true for VAEs. Can you explain why "z_i are independent conditioned by x" is true for VAEs?

---

> > > > > > > > > > > > > ### Author Response · Authors · 2019-11-12
> > > > > > > > > > > > > **Answers to Reviewer #2**
> > > > > > > > > > > > >
> > > > > > > > > > > > > Thank you for your consecutive reviews!
> > > > > > > > > > > > >
> > > > > > > > > > > > > As we mentioned, by definition, in that space, a latent variable covers only one generative factor and does not affect each other and we interpreted it as independence.
> > > > > > > > > > > > >
> > > > > > > > > > > > > In many existing disentangled representations, it is confirmed that even for the same data x, different z_i changes individual characteristics (e.g. background color, gender, etc.) that do not affect each other.

---

### Author Response · Authors · 2019-09-27
**Formatting error causing inconvenience to read**

Apologies to the readers - we identified a formatting error in the first paragraph of Section 3.1. Theorem 1 and Lemma 1 have been not written separately, but together in the main text.

---

### Decision · Program_Chairs · 2019-12-19

**Decision:**

Reject

**Comment:**

The paper proposes a new way to learn a disentangled representation by embedding the latent representation z into an explicit learnt orthogonal basis M. While the paper proposes an interesting new approach to disentangling, the reviewers agreed that it would benefit from further work in order to be accepted. In particular, after an extensive discussion it was still not clear whether the assumptions of Theorem 1 applied to VAEs, and whether Theorem 1 was necessary at all. In terms of experimental results, the discussions revealed that the method used supervision during training, while the baselines in the paper are all unsupervised. The authors are encouraged to add supervised baselines in the next iteration of the manuscript. For these reasons I recommend rejection.